# Host Shaping Associated Microbiota in Hydrothermal Vent Snails from the Indian Ocean Ridge

**DOI:** 10.3390/biology14080954

**Published:** 2025-07-29

**Authors:** Xiang Zeng, Jianwei Chen, Guilin Liu, Yadong Zhou, Liping Wang, Yaolei Zhang, Shanshan Liu, Zongze Shao

**Affiliations:** 1Key Laboratory of Marine Biogenetic Resources, Third Institute of Oceanography, Ministry of Natural Resources, Xiamen 361005, China; liping84@genomics.cn; 2Faculty of Marine Biology, Xiamen Ocean Vocational College, Xiamen 361100, China; 3BGI-Qingdao, BGI-Shenzhen, Qingdao 266555, China; chenjianwei@genomics.cn (J.C.); liuguijin@genomics.cn (G.L.); zhangyaolei@genomics.cn (Y.Z.); liushanshan@genomics.cn (S.L.); 4Laboratory of Genomics and Molecular Biomedicine, Department of Biology, University of Copenhagen, Universitetsparken 13, 2100 Copenhagen, Denmark; 5Key Laboratory of Marine Ecosystem and Biogeochemistry, Second Institute of Oceanography, Ministry of Natural Resources, Hangzhou 310012, China; yadong_zhou@sio.org.cn; 6Hainan Deep-Sea Technology Laboratory, Institution of Deep-Sea Life Sciences, IDSSE-BGI, IDSTI-CAS, Sanya 572000, China

**Keywords:** hydrothermal vents, snail, symbiosis, pyrosequencing, metagenome, metagenome-assembled genomes, ectosymbiont, endosymbiont

## Abstract

Snails are a dominant member of the fauna in hydrothermal vents in terms of abundance and biomass. Two species of hydrothermal vent snails, *Chrysomallon squamiferum* (scale snail) and *Gigantopelta aegis* (scaleless snail), are found along the Indian Ocean Ridge and inhabit areas of low-temperature hydrothermal diffuse flow. Although it is generally assumed that they rely on autotrophic endosymbionts to utilize organic substrates and energy for growth, little is known about their associations with microbiota, particularly regarding community structure, niche acclimation, and metabolic potential. Here, we focused on the diversity and roles of bacterial ectosymbionts on snail feet and endosymbionts in digestive glands of these two species to understand the role of symbiotic microbiota in niche adaptation of the host to harsh hydrothermal environments with steep physical and chemical gradients.

## 1. Introduction

Symbioses between animal groups and chemosynthetic bacteria occur in a wide range of habitats worldwide [1]. In deep-sea chemoautosynthetic ecosystems such as hydrothermal ecosystems, vent fauna thrives in dense populations, largely through symbiotic relationships with bacteria [2]. Environmental factors shape the free-living microbial community structure in hydrothermal vents [3,4]. However, we know little about how vent animals acquire their symbionts from the surrounding environment. This acquisition is a critical life-history step, generally following one of two pathways: vertical transmission, where symbionts are passed directly from parent to offspring, or horizontal transmission, where hosts acquire their symbionts anew from a free-living population in the environment with each generation [5]. Understanding the transmission mode is key to deciphering how these symbioses are established and maintained. The resulting holobiont (host plus its microbiota) then populates a specific niche, suggesting that these partnerships promote the successful propagation of hosts and lead to distinct habitat-utilization patterns [6,7].

Three species of hydrothermal snail, *Alviniconcha*, co-occur at several vent localities in the Eastern Lau Spreading Center (ELSC), which harbor different lineages of bacterial symbionts, such as *Alviniconcha boucheti*, associated with Campylobacteria, and *Alviniconcha kojimai* and *Alviniconcha strummeri*, which harbor two distinct lineages of Gammaproteobacteria (γ-1 and γ-lau) [7]. Their niche separation is presumably ascribed to variations in environmental parameters such as oxygen and sulfide concentrations [8]. In contrast, several chemosymbiotic bivalve species in the Lucinidae family are distributed across the globe and are all associated with a single cosmopolitan bacterial symbiont [9]. To date, the interaction between niche, host, and symbiont is still unclear, especially for hydrothermal animals.

Snails are an important component of the fauna in hydrothermal vents in terms of abundance and biomass [10]. The scaly-foot snail, *Chrysomallons quamiferum*, is found only at hydrothermal vents at ~3000 m depths in the Indian Ocean. The scaly-foot snail has two varieties without detectable genetic differences: black scaly-foot snails that were found in the Kairei field on the Central Indian Ridge and the Longqi field on the Southwest Indian Ridge (abbreviated as SWIR) and white scaly-foot individual snails that were found in the Solitaire field on the Central Indian Ridge [10] and the Wocan field on the Carlsberg Ridge (abbreviated as CR) of the Northwest Indian Ocean [11]. The color difference of scaly-foot snails may be due to the iron in the surrounding water from the endmember hydrothermal fluids by forming a greigite—an iron sulfide mineral covering the exterior of the black scaly-foot snail [12]. The snail of *Gigantopelta* without scales on its foot inhabits mainly the Indian Ocean and Southern Ocean, including two species, *Gigantopelta chessoia* found in the East Scotia Ridge and *Gigantopelta aegis* found in SWIR [10]. Genome analysis of the hydrothermal snails *Chrysomallon* and *Gigantopelta* found that these 2 deep-sea snails evolved independently, and their divergence from each other occurred ∼66.3 million years ago [11]. Scaly-foot snail *C. squamiferum* revealed significant enrichment for metabolism in the glycolysis pathway and citrate cycle (TCA cycle). In the *G. aegis* genome, we found unique and expanding histocompatibility complex (MHC) genes for immunity [11]. Both the *C. squamiferum* and *G. aegis* host genomes lack the biosynthetic capabilities of the same nutrients, including six amino acids, six vitamins, and coenzyme A, which could be supplied by symbionts [13]. These two genera (*Chrysomallon* and *Gigantopelta*) belong to the family Peltospiridae and presumably rely on thioautotrophic endosymbionts living in bacteriocytes inside the esophageal gland of the snail [14]. Previous studies have found that in scaly-foot snails, a dense population of Gammaproteobacteria populates the gland cells, and Epsilon- and Deltaproteobacteria colonize the scale surface, as determined by a 16S rDNA clone library [15]; additionally, metagenomic analysis revealed that one symbiont genome belongs to the Gammaproteobacteria (order Chromatiales) [16]. Lan et al. described two endosymbionts of Gammaproteobacteria MAGs of *G. aegis* [13].

At present, knowledge of the relationship between symbiosis and vent snails is limited. In this study, we compared black scaly-foot and white scaly-foot snails of *Chrysomallon squamiferum* and *Gigantopelta aegis* in the diversity and metabolism of epi- and endosymbionts. Insights into the associations of symbiont microbiota with vent snails will help us to better understand host adaptation to hydrothermal vent environments and the mechanisms underlying niche–host–symbiont interactions.

## 2. Materials and Methods

### 2.1. Sample Collection

Individuals of *Gigantopelta aegis* (abbreviated as G) and the black scaly-foot snail *Chrysomallon squamiferum* (abbreviated as BC) were collected from the Longqi hydrothermal field (49.9° E, 37.8° S, 2780 m depth) on the SWIR in March 2015 during the Chinese DY35th cruise of R/V Xiangyanghong 9. The white scaly-foot snail *Chrysomallon squamiferum* (abbreviated as WC) was from the Wocan hydrothermal field (60.5° E, 6.4° N, 2926 m depth) on the CR in March 2017 during the Chinese DY38th cruise of R/V Xiangyanghong 9 (Figure 1; Appendix A). Using the port manipulator of the Jiaolong HOV submersible, snail samples were recovered in an insulated container and delivered to the surface. On board, samples were immediately thoroughly washed with sterile seawater and frozen in liquid nitrogen. Frozen tissues, including foot tissue (abbreviated as F) and esophageal gland (abbreviated as G), from 9 snails were weighed, washed, ground in liquid nitrogen, and used to extract DNA using a DNeasy Blood and Tissue Kit (Qiagen, Hilden, Germany) according to the manufacturer’s instructions (Appendix A).

Two scaly sclerites from *C. squamiferum* were examined for the presence of bacterial ectosymbionts. Samples for scanning electron microscopy were dehydrated in 70% ethanol and then air-dried and examined under an FEI Quanta 450 Field Emission Scanning Electron Microscope (Hillsboro, OR, USA) equipped with Quorum PP3000T (Quorum Technologies Ltd., Laughton, UK). Temperature and salinity of the fluids surrounding the snails were measured in situ using electrochemical sensors deployed by a manipulator arm (HOV). The fluid was filtered for pH determination on board at room temperature by a pH meter. The concentrations of CH_4_, H_2_, and H_2_S in the fluid samples were detected by gas chromatography [17].

### 2.2. 16S rRNA Gene Library Preparation, Sequencing and Analysis for 17 Samples from C. squamiferum (BC, WC) and G. aegis (G)

The concentration of the DNA was determined with a NanoDrop ND-2000 (Thermo Scientific, Wilmington, NC, USA). The V4 regions of the 16S rRNA gene were amplified from each sample using universal prokaryotic primers (515F: GTGCCAGCMGCCGCGGTAA, 806R: GGACTACHVGGGTWTCTAAT) with barcodes [18]. The PCR products were mixed in equal amounts, and the target bank was gel-cut by electrophoresis in a 2% agarose gel. Then, a single-stranded circular DNA library was generated. After assessment with a Qubit@2.0 Fluorometer (Thermo Scientific, USA) and Agilent Bioanalyzer 2100 system, the library was sequenced on the MGISEQ-2000 platform at BGI-Qingdao (Qingdao, China), and raw paired-end sequencing reads (2 × 150 bp) were generated.

The adapter contamination and low-quality reads of raw data were filtered out by SOAPnuke (v1.5.6). The paired-end clean reads were combined into tags by FLASH (v1.2.11), and then the denoising ASVs were generated by USEARCH (v10.0.240) using the unoise3 algorithm with the parameter “-minsize 8” to obtain the ASV abundance profiles [19]. The ASV taxonomic assignment was carried out by RDP-Classifier (v2.2) against the Greengene (v201305) database with a 0.8 confidence cutoff for microbial community structure analysis. The alpha diversity index, including observed ASVs, Shannon indices, Chao indices, and Bray_CurtisBeta diversity distance and principal coordinate analysis (PCoA), was analyzed using the microbial ecology software package QIIME (v1.9.1) [20]. All significant analyses, including the Shannon–Wiener index, species abundance, and ASV abundance, were determined by the Wilcoxon test using R software (v3.4.1).

### 2.3. Metagenomic Sequencing, Assembly and Binning for Six Samples from C. squamiferum (BC, WC) and G. aegis (G)

DNA was sheared to 400–600 bp using a Covaris S-series sonicator, and metagenomic library construction was completed using the MGIEasy DNA Rapid Library Prep Kit (MGI-Shenzhen, catalog no. 1000006985, China) following the manufacturer’s instructions. Metagenomic sequencing was performed on six samples: two snail foot metagenomes from *C. squamiferum* (WC3F, BC1F) and two gland metagenomes from *C. squamiferum* (WC1G, BC1G). Additionally, two snail foot metagenomes were analyzed from *G. aegis* (G1F, G3F). Despite repeated attempts, metagenomes could not be successfully constructed from the gland tissues of *G. aegis*. Refer to Appendix A for sample details (Appendix A). Metagenomic sequencing was performed on BGISEQ-500 platforms at BGI-Qingdao (Qingdao, China) in a 100 bp paired-end read model. After filtering low-quality data, duplication reads, adapter contamination reads, and host genome sequences by KneadData (v0.7.2, http://huttenhower.sph.harvard.edu/kneaddata, accessed on 5 January 2023), we obtained high-quality data for all foot and gland metagenomics samples. The metagenomics data were assembled using idbaud (v1.1.2) [21] with the parameters “--mink 23 --maxk 83 --step 20”, discarding contigs smaller than 300 bp.

Subsequent binning analyses were performed in a supervised fashion using both tetranucleotide frequency and coverage for clustering by MetaWRAP (v1.1.5) [22]. The “Binning” module using methods metabat2, concoct, and maxbin2 and the “Bin_refinement” module with parameters “contamination < 5% and completeness > 80%” were used to generate the metagenome-assembled genomes (MAGs) for every assembled genome. We used GTDBsoftware (v2.3.2) [23] to confirm the taxonomic assignment of the identified MAGs. The completeness and quality of the final MAGs were assessed by CheckM (v1.0.7) [24].

### 2.4. Gene Annotation and Metabolic Analysis of MAGs and Metagenomes

To distinguish microbial contigs from host-derived sequences within the metagenomic assemblies, all assembled contigs were aligned against the published host genomes [11] using BLASTn(v2.13.0). Contigs with high sequence similarity (e.g., >95% identity over >80% of the contig length) to the host genome were classified as host contamination and removed. We also examined the GC content of the remaining contigs, as symbiont genomes often exhibit a distinct GC percentage from their host.

Genes were identified using Glimmer (v3.02) for MAGs and MetaGenemark (v3.26) for metagenomics assembly contigs, followed by manual screening. The genes from MAGs and the nonredundant gene set were compared with the nonredundant protein database of the NCBI (Nr) (v20200204) and Kyoto Encyclopedia of Genes and Genomes (KEGG) (v87.0) databases using BLASTP with an e-value < 1 × 10^−5^. The metabolic pathway was reconstructed using the KEGG [25] and MetaCyc [26] databases. Gene prediction and annotation of MAGs were also performed with RASTtk [27]. Hydrogenase and iron oxidase/reductase analyses were further analyzed by HydDB [28] and FeGenie [29]. Additionally, all metagenomics gene sets were merged, and the nonredundant gene set was constructed by CD-Hit (v4.6.6) with 95% identity. To determine the differential abundance of functional features between different snails, Wilcoxon-test Metastats analysis was applied, and the heatmap of significant KO abundance was drawn by the R “pheatmap” package.

### 2.5. Phylogenomic Analysis of MAGs

Marker genes of MAGs and reference genomes were classified, and consensus alignment sequences were generated according to the latest version of GTDB [23]. RNAmmer (v1.2) [30] was used to retrieve 16S rRNA sequences from metagenomic assemblies, and all predicted and reference 16S rRNA sequences were aligned by the QIIME (v1.9.1) software package “align_seqs.py” using the PyNAST method. The phylogenetic trees of 16S rRNA and marker genes were constructed by FastTree in the “phyml” model. Average nucleotide identities (ANIs) to the next sequenced relative and between the assemblies were calculated using OrthoANI software (v0.6.0) [31].

### 2.6. Data Availability

The 16S rRNA genes and metagenomics sequencing data that support the findings of this study have been deposited in the CNSA (https://db.cngb.org/cnsa/, accessed on 13 October 2021) of CNGBdb with accession code CNP0001245. The assembled and annotated symbiont genomes are also publicly available on the RAST server (http://rast.theseed.org/, accessed on 10 May 2022) using the guest login with IDs 6666666.654542-6666666.654548, 6666666.654559, and 6666666.654594-6666666.654598.

## 3. Results

### 3.1. Distributions of Snails on Indian Ocean Ridges and Ultrastructural Characterization of Scaly-Foot Snails by Scanning Electron Microscopy (SEM)

In the Longqi hydrothermal field of the SWIR ridge, black scaly-foot *C. squamiferum* (BC) and *G. aegis* (G) colonize together in dense populations on vent chimney basals close to visible diffuse flow of vent fluid, and at sampling site DFF11, the vent chimney basals are occupied solely by these two species (Figure 1C). The maximum temperature at these diffusing areas was 13.3 °C. The diffuse flow has a high concentration of methane (12.68 mM) and hydrogen (8.197 mM). The white scaly-foot *C. squamiferum* inhabits the collapsed sulfide chimney with low-temperature diffusing flow around the high-temperature black smoking vent of the Carlsberg Ridge of the Northwest Indian Ocean in the hydrothermal field of Wocan (Figure 1B). As previously reported, *Gigantopelta aegis* has not been found in CR [32].

Scanning electron microscopy revealed the difference in sclerites between white (WC) and black scaly-foot snails (BC) of *C. squamiferum*. On the foot of WC, the sclerites were colonized by long microbial filaments with short rods attached to the filament surface, while on the sclerites of the BC snail, iron sulfide accumulation was observed sticking to cocci- and filamentous bacteria (Appendix A).

### 3.2. Characterization of Microbial Communities Associated with Hydrothermal Snails

#### 3.2.1. Microbial Community Structure Revealed by High Throughput Sequencing

Seventeen 16S rRNA gene amplicon libraries (three individuals per sample) were constructed from the foot and gland of *C. squamiferum* and *G. aegis* (Table 1). Raw sequencing reads were obtained from these libraries on the MGISEQ-2000 platform. After merging and processing the raw reads, 1,265,697 filtered tags with an average length of >252 bp of the 16S rRNA gene spanning the variable region V4 were clustered into denoising ASVs (also called zero-radius ASVs, or ZOTUs (Operational Taxonomic Units)) at 100% sequence similarity, of which 34.72%, 28.14%, and 37.14% were from the three individuals of BC, WC, and G, respectively (Table 1). A total of 1014 ASVs were obtained. Nearly all sequence tags (99.9982%) were assigned to the bacterial domain. Rarefaction curves approached a plateau (Appendix A), which indicated that sequences adequately represented bacterial composition in snail samples. Rank-abundance curves showed that the scaly-foot snail *C. squamiferum*, especially gland samples, contained more diverse sequence tags and relatively lower abundance than *G. aegis* (Appendix A).

The diversity indices, including the Shannon, Simpson, and community richness Modified itModified itChao1 and ACE indices, all revealed microbiota variations between *C. squamiferum* and *G. aegis* and between the scaly-foot and gland samples in *C. squamiferum* (Table 1). The Shannon index of snail *C. squamiferum* samples ranged from 1.32 to 4.60, whereas the Shannon index of the *G. aegis* samples ranged from 0.14 to 0.18. The Chao1 index ranged from 202.79 to 508.00 in *C. squamiferum* samples and from 140.25 to 246.86 in *G. aegis*. This result indicated that obviously higher bacterial diversity occurred in *C. squamiferum* than in *G. aegis*. For *C. squamiferum*, the diversity of WC was higher than the diversity of BC (Table 1). The mean Shannon indices of bacterial communities in the glands (2.49) were slightly higher than the mean Shannon indices on the foot (2.17). Likewise, the mean Chao1 index in glands (341.04) was higher than the mean Chao1 index in the foot (284.07).

Then, PCoA analysis (Appendix A) and the Bray–Curtis cluster tree (Appendix A) based on 16S rRNA gene ASVs were performed to examine the differences in bacterial communities. Two principal components (PC1 41.25% and PC2 18.83%) explained 60.08% of the total variation in the bacterial community in the samples. Both BC and WC were obviously separated from *G. aegis* along axis PC1. The minor factor PC2 impacted the communities of BC and WC. The BC and WC samples had 455 identical symbiont ASVs, which accounted for ~96.36% and ~80.53% of the ASV abundance of the two species, respectively (Appendix A). Only 237 ASVs were shared among the three snails, accounting for ~83.64% of BC samples, ~48.39% of WC samples, and 99.46% of G samples (Appendix A). The Bray–Curtis cluster tree (Appendix A) indicated that 17 populations could be divided into two major clusters: Cluster 1 included mainly the populations of *G. aegis* (GG and GF), with Cluster 2 consisting of the remaining 12 populations of *C. squamiferum* (C), which were further divided into two short branches in accordance with snail hosts of white (WC) and black (BC). The results implied that the microbial community structure of these snails was relevant mainly to the host and less relevant to the adjacent geochemistry of the habitat environments.

The microbial community structures revealed striking and distinct patterns across the two snail species and between the *C. squamiferum* ecotypes (Figure 2a,b).

The black scaly-foot snail (BC) displayed a stable epibiotic community with a variable endosymbiotic core. Its foot surface (epibionts) was characterized by a low-diversity community overwhelmingly dominated by the phylum Campylobacterota (67.01–83.57%). Specifically, the family Helicobacteraceae, driven by key taxa such as ASV0003, was the principal component. This is consistent with previous findings that genera in this family are key sulfur-oxidizing bacteria (SOB) in hydrothermal environments [8]. In contrast, the gland community (endobionts) was more variable. Some individuals were dominated by Gammaproteobacteria, with a single taxon from the order Chromatiales (ASV0004) comprising over 74% of the community, while others hosted a more mixed assemblage of Campylobacterota, Alphaproteobacteria, Bacteroidetes, and Firmicutes (Figure 2a).

In contrast, the white scaly-foot snail (WC) harbored a significantly more diverse and heterogeneous microbiome with high inter-individual variability. The strong dominance of a single Campylobacterota lineage seen on the black ecotype was absent. Instead, WC communities comprised a shifting mixture of several major phyla, including Gammaproteobacteria, Bacteroidetes, Deltaproteobacteria, Alphaproteobacteria, and Firmicutes. The dominant taxa frequently differed not only between individuals but also between the foot and gland within the same animal, indicating a less specialized microbial association.

*G. aegis* presented a highly specialized gammaproteobacterial symbiosis. Both its foot and gland communities were almost entirely composed of Gammaproteobacteria, with the order Thiotrichales, represented almost entirely by a single amplicon sequence variant (ASV0001), accounting for over 96% of total sequences (Figure 2). Genera within Thiotrichales are known to thrive in vent environments [33]. This Thiotrichales-dominated symbiosis is notably different from the Chromatiales-based symbioses reported in other vent fauna [7,34,35].

In summary, these snails exhibit three distinct symbiotic signatures. *G. aegis* relies on a highly specialized Thiotrichales (Gammaproteobacteria) symbiosis. *C. squamiferum*, on the other hand, hosts a more complex suite of symbionts, with the black ecotype (BC) maintaining a consistent Campylobacterota-dominated epibiome, while the white ecotype (WC) is characterized by a more generalized and variable microbial community. The remarkably lower concentrations of methane and hydrogen in the Wocan field vent fluids compared to those in the Longqi field (DFF11) of SWIR (Appendix A), coupled with the relatively low H_2_S concentrations (1400–2200 nM) compared to other hydrothermal plumes, likely results in reduced sulfide availability. This limitation may be a key factor in the reduced colonization of the white scaly-foot snail scales by hydrogen- and sulfur-oxidizing *Campylobacterota*.

#### 3.2.2. Metabolic Potential of the Microbial Community Based on Metagenomic Analysis

We sequenced and analyzed the metagenomes of six samples from white scaly-foot *C. squamiferum* (WC3F, WC1G), black scaly-foot *C. squamiferum* (BC1F, BC1G), and *G. aegis* (G1F, G3F) (Appendix A). Unfortunately, the gland metagenome from *G. aegis* was not constructed.

The functional capacity was determined according to the annotation of ORFs predicted from the assembled contigs. The nonredundant gene catalog containing a total of 782,337 ORFs was constructed with an average length of 329.57 bp. The nonredundant predicted genes were then aligned with the KEGG gene database. We identified a total of 195,997 KEGG genes and assigned them to 426 KEGG pathways. PCA based on KEGG annotations of metagenomic data showed obvious differential distributions of KEGG pathways between *C. squamiferum* and *G. aegis* (Appendix A). A heatmap of functional genes based on the KEGG pathway annotation also shows differences between *C. squamiferum* and *G. aegis* (Figure 3). The genes that were highly enriched in *G. aegis* (G1F, G3F) were associated with the sulfur cycle and methane-oxidizing pathway, indicated by the corresponding key genes encoding dissimilatory sulfite reductase (DsrAB, K11180, K11181), adenylylsulfate reductase (AprAB, K00394, K00395), sulfide dehydrogenase (fccAB, K017229, K017230), sulfur-oxidizing protein (SoxBYZ, K017224, K017226, K017227), and methane/ammonia monooxygenase (pmoABC, K10944–10946). Compared with *G.aegis,* the metabolic pathways in *C. squamiferum* were enriched with carbon fixation (WL and rTCA), denitrification and glycolysis in addition to sulfur cycle genes (Figure 3), reflected by the corresponding key genes encoding acetyl-CoA synthase (K00194, K00197, K14138), anaerobic carbon-monoxide dehydrogenase (K00196, K00198) in WL pathway, ATP-citrate lyase (aclAB, K15230, K15231) in reductive TCA cycle, nitrous-oxide reductase (nos. DLZ, K07218, K00376, K19342) in denitrification (N_2_-forming), menaquinol-cytochrome c reductase subunit NrfD (K04015) in dissimilatory nitrate reduction, pyruvate ferredoxin oxidoreductase (PFOR, K00169) and butyrate kinase (K00929) to catalyze the oxidative decarboxylation of pyruvate to acetyl-CoA and produce butyrate, cytochrome d ubiquinol oxidase, subunit II (K00426) in oxygen utilization. Comparing *C. squamiferum* and *G. aegis* revealed that they not only host different bacterial symbionts in microbial taxonomic profiles but also significantly vary in metabolic potential.

#### 3.2.3. Phylogenomics and Predicted Metabolic Capabilities of Dominant Metagenome-Assembled Genomes (MAGs)

After filtration of low-quality MAGs, 13 MAGs with completeness ≥80% and contamination ≤ 5% were obtained for further analysis (Table 2). The 13 retrieved high-quality MAGs were taxonomically assigned to three phyla, including *Gammaproteobacteria* (8 MAGs), *Campylobacterota* (4 MAGs), and *Bacteroidota* (1 MAG). Additionally, two MAGs affiliated with *Firmicutes* and *Bacteroidota* from sample BC1F were also binned out but not analyzed due to their low completeness (71.54% and 60.84%).

Phylogenomic analysis based on 120 marker genes was performed between 13 MAGs and their related genomes, including symbionts from marine animals (Figure 4). The phylogenomic tree of 13 MAGs indicated the presence of multiple symbiont phylotypes of hydrothermal snails from the Indian Ocean in this study. Four distinct lineages in the class *Gammaproteobacteria* were identified, including four MAGs of the order *Chromatiales*, two of the order *Thiotrichales*, one in the order *Methylococcales*, and one unclassified group distinct from the other lineages. Four MAGs from scaly snails were affiliated with the *Sulfurovum* genus of *Campylobacterota*, and one MAG belonged to the phylum *Bacteroidota*.

To understand the functional roles of the dominant symbionts, we focused on their core metabolic pathways for carbon fixation, energy generation, and key adaptations to the vent environment (Figure 5, Table 2).

##### Chromatiales of Gammaproteobacteria

The order Chromatiales is frequently found as endosymbionts in hydrothermal vent fauna, including snails, tube worms, and sponges [13,35,36,37]. We recovered four Chromatiales MAGs that segregated into two distinct families. The first group, comprising endosymbionts from the glands of both black (BC1G.bin.1) and white (WC1G.bin.1) *C. squamiferum*, belonged to the family Chromatiaceae. These two MAGs were nearly identical to a previously reported *C. squamiferum* endosymbiont from the Kairei field (98.5% ANI) [36], confirming a stable host–symbiont association across different Indian Ocean vent fields. Metabolically, they are versatile chemolithoautotrophs, fixing carbon via the Calvin–Benson–Bassham (CBB) cycle using both Form I and Form II RubisCO, which suggests an ability to adapt to varying CO_2_/O_2_ conditions [38]. They possess an extensive repertoire of sulfur-oxidizing pathways (Sox, Hdr, reverse Dsr), genes for hydrogen utilization, and mechanisms for heavy metal detoxification as reported [39,40,41,42]. The second group, recovered from the foot of *G. aegis* (G3F.bin1, G1F.bin2), belonged to the family Ectothiorhodospiraceae and was phylogenetically distinct from an *Alviniconcha* symbiont lineage (γ-1) [43] (Figure 4). These epibionts are also sulfur-oxidizing autotrophs but primarily utilize Form II RubisCO for carbon fixation via the CBB cycle as reported [44,45,46,47,48].

##### Thiotrichales of Gammaproteobacteria

Two epibiotic MAGs from the white scaly-foot snail (WC3F.bin.10, WC3F.bin.17) were affiliated with filamentous sulfur-oxidizers in the order Thiotrichales (e.g., *Thiothrix*, *Leucothrix*), which have been described on other vent organisms [49]. These MAGs encode a mixotrophic metabolism, using the CBB cycle for carbon fixation and oxidizing sulfide via sulfide dehydrogenase (FccAB), a pathway efficient in low-sulfide conditions. Notably, both MAGs possess *cyc2* genes (cyc2-Cluster 3), suggesting a previously unreported capacity for iron oxidation among these symbionts [50,51,52] (Appendix A).

##### Methylococcales of Gammaproteobacteria

A methanotrophic MAG (G1F.bin.1) was recovered from *G. aegis* and is phylogenetically close to *Methylomarinum vadi*, a methanotroph from a shallow hydrothermal system [53,54]. Symbiotic Methylococcales are also known from marine mussels and sponges [55]. This MAG harbors the complete genetic toolkit for aerobic methane oxidation, including methane monooxygenase (*pmoCAB*), and assimilates C1 compounds via the ribulose monophosphate (RuMP) pathway (Figure 5D). Its genome also contains genes for hydrogen utilization and glycogen storage, key adaptations for a symbiotic lifestyle [56].

##### *Candidatus* Endothiobacterales of Gammaproteobacteria

A MAG from the white scaly-foot snail gland (WC1G.bin2) represents a previously uncharacterized lineage of Gammaproteobacteria. Phylogenomically, it is highly distinct, clustering only with a sulfur-oxidizing symbiont from a deep-sea glass sponge [57] and sharing low ANI values (<68.3%) with other known orders (Figure 4). Based on this evidence, we propose the new candidate order “*Candidatus* Endothiobacterales”. Metabolically, this MAG is a sulfur-oxidizing autotroph, utilizing the CBB cycle (Form II RubisCO) for carbon fixation and possessing a comprehensive suite of sulfur oxidation pathways (Figure 5C).

##### Campylobacterales of Campylobacterota

Four MAGs, primarily from the foot surface of *C. squamiferum*, were classified within the genus *Sulfurovum*, a well-known chemosynthetic group in vent environments where its members thrive attached to surfaces [8,58]. These symbionts are chemolithoautotrophs that fix carbon via the reductive tricarboxylic acid (rTCA) cycle (Figure 5E). They oxidize sulfur compounds using sulfide-quinone reductase (Sqr) and the Sox system. Consistent with a symbiotic lifestyle, their genomes were significantly smaller (1.38–1.61 Mbp) than those of their free-living relatives (Appendix A) [59,60,61].

##### Flavobacteriales of Bacteroidetes

A single MAG from the white scaly-foot snail (WC3F.bin18) belongs to the order Flavobacteriales (phylum Bacteroidetes). This MAG exhibits features of a heterotrophic, surface-associated lifestyle, a common strategy for this phylum, which often degrades polymers on particles [62]. Its genome encodes numerous carbohydrate-active enzymes (CAZymes) for degrading complex organic matter, as well as genes for gliding motility (Figure 5F). This suggests a role in utilizing organic polymers produced by the primary chemosynthetic symbionts on the snail’s scales, a niche also observed for Bacteroidetes ectobionts on *Sulfurovum* filaments [63].

SOX, sulfur-oxidizing protein; SQR, sulfide:quinone oxidoreductase; FCC, flavocytochrome-c sulfide dehydrogenase; DsrEF intracellular sulfur oxidation protein; HdrABC, heterodisulfide reductase; DsrAB, dissimilatory sulfite reductase, subunit AB; AprAB, adenylylsulfate reductase, subunit AB; Sat, sulfate adenylyltransferase; SoeABC, membrane-bound sulfite dehydrogenase; MBHL, membrane-bound hydrogenase; Hox, cytoplasmic NiFe hydrogenases; pMMO methane monooxygenase; CODH, carbon monoxide dehydrogenase; Cyc2, cyctochrome C oxidase; ArsC, arsenate reductase; MerA, mercury reductase; Cox, cytochrome c oxidase; Cco, cytochrome c oxidoreductase; Glc, glycolate oxidase; NapAB, periplasmic nitrate reductase; NasA, Nitrate reductase; NirBD, nitrite reductase; Nuo, NADH ubiquinone oxidoreductase; Fdh, formate dehydrogenase; Sdh, succinate dehydrogenase; Nqr, sodium-dependent NADH dehydrogenase.

## 4. Discussion

Nutritional symbioses between eukaryotic organisms and autotrophic microbes are ubiquitous throughout the oceans of the Earth. These associations have allowed marine organisms to flourish in nutrient-limited or extreme environments where they reach population densities unmatched by their nonsymbiotic relatives [64]. In this study, the diversity and host specificity of host–microbe relationships in hydrothermal snails were analyzed by comparing the microbiota of different snail species in the same niche and two varieties of the same species in different hydrothermal inhabitants (Figure 6).

CBB, Calvin–Benson–Bassham cycle; rTCA, the reductive tricarboxylic acid cycle; WL, reductive acetyl-CoA pathway (Wood-Ljungdahl pathway); SOX, sulfur-oxidizing protein; SQR, sulfide:quinone oxidoreductase; FCC, flavocytochrome-c sulfide dehydrogenase; DsrEF Intracellular sulfur oxidation protein; HdrABC, heterodisulfide reductase; DsrAB, dissimilatory sulfite reductase, subunit AB; AprAB, adenylylsulfate reductase, subunit AB; Sat, sulfate adenylyltransferase; SoeABC, membrane-bound sulfite dehydrogenase; MBHL, membrane-bound hydrogenase; Hox, cytoplasmic NiFe hydrogenases; pMMO, methane monooxygenase; CODH, carbon monoxide dehydrogenase; Cyc2, cytochrome C oxidase; ArsC, arsenate reductase; MerA, mercury reductase.

### 4.1. Host-Specific Symbiont Type in Hydrothermal Snails

The hydrothermal snail *G. aegis* and the scaly-foot snail *C. squamiferum* were affiliated with the same family (*Peltospiridae*), restricted to chemosynthetic ecosystems [10], and phylogenetically independent of other shallow-water gastropods [11]. In this study, these two snail species, *C. squamiferum* and *G. aegis*, live together in the same niche in the Longqi vent, SWIR. *G. aegis* harbors extremely overabundant *Gammaproteobacteria*, belonging mainly to the order *Chromatiales* and *Methyloccales*. As reported for two MAGs in *G. aegis* [13], G1F.bin2 was close to abundant sulfur-oxidizing bacteria, and G1F.bin1 was close to the less common methane-oxidizing bacteria. Another sulfur-oxidizing bacterium, G3F.bin1, which was close to the filamentous SOB *Beggiatoa alba*, has never been reported before. Compared with SOB G1F.bin2, G3F.bin1 can use hydrogen and sulfur species as electron donors but cannot produce pyridoxine (vitamin B6) and cobalamin (vitamin B12). Based on the low bacterial diversity in *G. aegi*, we assumed that powerful host-immune responses and disturbance may contribute to it according to host genome analysis. The black-scale snail *C. squamiferum* resides on an active chimney neighboring *G. aegi* but houses *Campylobacterota* and *Gammaproteobacteria* as dominant members in the symbiotic community, in addition to diverse bacteria of *Delta-Alphaproteobacteria*, *Bacteroidetes*, and *Firmicutes*. Their symbiotic bacteria can use versatile energy sources by H_2_ oxidation, sulfur oxidation, and iron oxidation coupled with the reduction of sulfur and multiple oxygen receptors. The host *C. squamiferum* also displayed significant enrichment of metabolism in the glycolysis pathway and citrate cycle (TCA cycle) [11], which helped to maintain the levels of ATP and metabolic pathways to provide additional nutrients for diverse symbionts. Therefore, the metabolic versatility of the microbiome, along with other key life-history traits such as dispersal potential, likely contributes to the broader distribution of *C. squamiferum* compared to *G. aegis* along the Indian Ocean Ridge.

In summary, even when living in the same niche on a hydrothermal vent chimney, different snail species harbor totally different symbionts. The phylosymbiosis found in the hydrothermal gastropod family *Peltospiridae* (this study) and chemosymbiotic bivalve family *Lucinidae* [9] implied that it could be a trait in some chemosymbiotic animals.

### 4.2. “Core” Microbial Community with Abundance Variation in the Hydrothermal Snail C. squamiferum

Black scaly-foot and white scaly-foot varieties of snail *C. squamiferum* spread across hydrothermal fields from SWIR to NWIR via CR in the Indian Ocean. As previously reported, the snail *C. squamiferum* can biologically control the mineralization of iron sulfide nanoparticles through channel-like columnar structures within the scales to enrich sulfur. Then, sulfur/iron diffusion further occurs in seawater [12]. Due to the low iron content in the fluid, the white scaly-foot snail without sulfides on the scale spread in the Solitaire Field in CIR Ridge [65] and Wocan Field in the Carlsberg Ridge (this study). Sulfur isotope analysis evidenced the accumulation of ^32^S in the scaly-foot snail by bacteria. In this study, black scaly-foot (BC) snails live in H_2_- and CH_4_-abundant sites, while white-scaly (WC) snails grow with depleted H_2_ and CH_4_ (Appendix A). We found that black scaling (BC) was obviously different from white scaling (WC) in the abundance of the bacterial community by amplicon sequencing (Figure 2) and microscopy analysis (Appendix A). Black scaly-foot hosts harbor mainly 67.01–83.57% *Campylobacterota* bacteria, among which the sulfur-oxidizing bacteria *Sulfurovum* spp. are the predominant key member. The predominance of *Sulfurovum* spp. occurs on the surfaces of vent animals such as snails and shrimp [15,66]. The bacteria associated with white scaly-foot WC were more diverse, composed of *Campylobacterota*, *Gammaproteobacteria*, *Deltaproteobacteria*, *Bacteroidetes*, and *Firmicutes* (Figure 2); intriguingly, WC contained a higher content of heterotrophic groups and less autotrophic *Campylobacterota*. Normally, the proportion of heterotrophs, including *Deltaproteobacteria* and *Bacteroidetes*, is greater in inactive hydrothermal sulfides than in active hydrothermal sulfides [67,68]. Firmicutes are also always found in nonhydrothermal sediments [69,70]. Deltaproteobacterial symbionts may also act as sulfate reducers on scales, indicating that the white scaly-foot snail *C. squamiferum* seems adaptive to inactive hydrothermal environments such as sulfides and sediments, in which reducing gases such as H_2_S, H_2_, and CH_4_ are not effluent but the content of organic carbons may be higher.

Although the two subtypes of scaly-foot snails of *C. squamiferum* are distributed across a large geographical distance, they contain the same sulfur-oxidizing endosymbiont, which formed a small cluster with high similarity (ANI value > 98.5%, Figure 3), including WC1G.bin.1 from white scaly-foot snails from the Wocan field in the Carlsberg Ridge, BC1G.bin.1 from black scaly-foot snails in the Longqi field of the Southwest Indian Ridge, and one black scaly-foot snail from the Kairei hydrothermal field in the central Indian Ocean [37], indicating that this endosymbiont belongs to the family *Chromatiaceae* in the order *Chromatiales* of *Gammaproteobacteria*, which tightly evolved with *C. squamiferum* and played an important role in symbiosis. This endosymbiont is another host-specific symbiont type identified in the hydrothermal gastropod family Peltospiridae. The transmission mode of this symbiont, whether vertical, horizontal, or mixed, requires further investigation. Lan et al. (2021) provide evidence suggesting a mixed transmission mode for *C. squamiferum* symbionts, highlighting the complexity of symbiont acquisition in these systems [13].

The “core” microbial community with key endosymbionts was transmittable among individuals of *C. squamiferum* spreading along the Indian Ocean Ridge, but variations in microbial abundance showed a microbial community in response to environmental stressors. The symbionts respond sensitively to the environment and have functions in nutrient supply and sulfide detoxification for the host, which is an important part of a complex adaptive system of *C. squamiferum*.

### 4.3. The Association of Hydrothermal Snails and Their Symbionts to Adapt to the Environment

The dominant macrofaunal species in hydrothermal vents are typically symbiotic primary consumers. Most chemosynthetic holobionts (host–symbiont associations) are sustained by energy metabolism of either sulfur or methane or hydrogen oxidation [1,13,71]. We compared the four studied species of hydrothermal chemosynthetic snails and their symbionts (Table 3). Snails *Gigantopelta* spp. and *Ifremeria nautilei* are distributed in a limited area, whereas *Alviniconcha* spp. and *Chrysomallon squamiferum* spread wider in the Pacific and Indian oceans (Table 3)*. Gammaproteobacteria*, especially the order *Chromatiales*, are the main and transmittable endosymbionts in these four species, which encoded carbon fixation and a hybrid Sox-reverse Dsr pathway, which would allow carbon fixation and the oxidation of thiosulfate, elemental sulfur, and sulfide as energy sources. *Campylobacterota* could be symbionts in the gills of *Alviniconcha* spp. (mainly *Sulfurimonas* and *Sulfurovum*) and on the scale of *C. squamiferum* (mainly *Sulfurovum*). They showed high similarity with free-living *Sulfurimonas* spp. and *Sulfurovum* spp., indicating their acquisition from environments. Compared with only the CBB cycle in *G. aegis* and *Ifremeria nautilei*, three major metabolic pathways for carbon fixation were found in symbiotic communities of hydrothermal snail *C. squamiferum*, including the CBB cycle, reductive tricarboxylic acid (rTCA) cycle, and reductive acetyl-CoA pathway (Wood-Ljungdahl pathway) (Figure 3 and Figure 5). A high proportion of heterotrophic bacteria in the gland of *C. squamiferum* are devoted to degrading and transporting various carbohydrates, amino acids, and peptides, suggesting a broad substrate spectrum for both carbon and nitrogen gain. Notably, most symbiont MAGs contain arsenate reductase (AsrC), mercury reductase (MerA), and copper resistance-related genes, indicating that those symbionts were adapted and helped the host develop metal resistance.

Collectively, these patterns in *Chrysomallon* might suggest that different host tissues represent distinct ecological niches that select for specific microbial partners. External surfaces, exposed to the dynamic chemistry of the surrounding water, favor a more diverse community adapted to environmental fluctuations, whereas stable, internal organs select for highly specialized endosymbionts optimized for nutrient provisioning. Although the metabolic flexibility of symbionts may contribute to the distribution and abundance of hydrothermal chemosynthetic snails, further research is needed to disentangle its effects from those of other factors, such as dispersal potential, habitat conditions, and interspecies competition.

## 5. Conclusions

The microbial symbionts of two snail species (black scale/white scale *Chrysomallon squamiferum* and *Gigantopelta aegis*) were characterized to understand their relationship with hydrothermal snails. Two snail species in the Indian Ocean harbor phylogenetically and functionally distant symbionts. *G. aegis* has strict selection on bacterial symbionts of sulfur-oxidizing bacteria of the family *Ectothiorhodospiraceae* and methane-oxidizing bacteria of the family *Methylococcaceae*. In contrast, snails of *C. squamiferum* adapt to a wider habitation range by recruiting a “core” community with changeable abundance in accordance with inhabitant spreading. The diverse “core” community includes *Campylobacterota*, *Gamma-*, *Delta-*, and *Alpha-Proteobacteria*, *Bacteroidetes*, and *Firmicutes*. These data highlight that the host largely shapes the associated microbiota in hydrothermal vent snails and that the local environment has a selective impact on the host.

## Figures and Tables

**Figure 1 biology-14-00954-f001:**
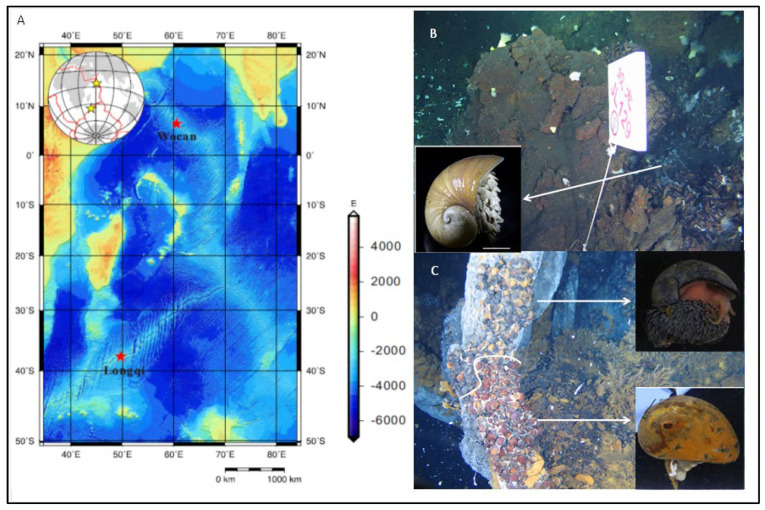
The sampling sites of snails living in the deep-sea hydrothermal field. (**A**) Map showing the sampling locations; (**B**) *Chrysomallon squamiferum* (white scaly) habit in Wocan Vent, NWIR; (**C**) Zonation of *Chrysomallon squamiferum* (black scaly) and *Gigantopelta aegis* in Longqi Vent, SWIR.

**Figure 2 biology-14-00954-f002:**
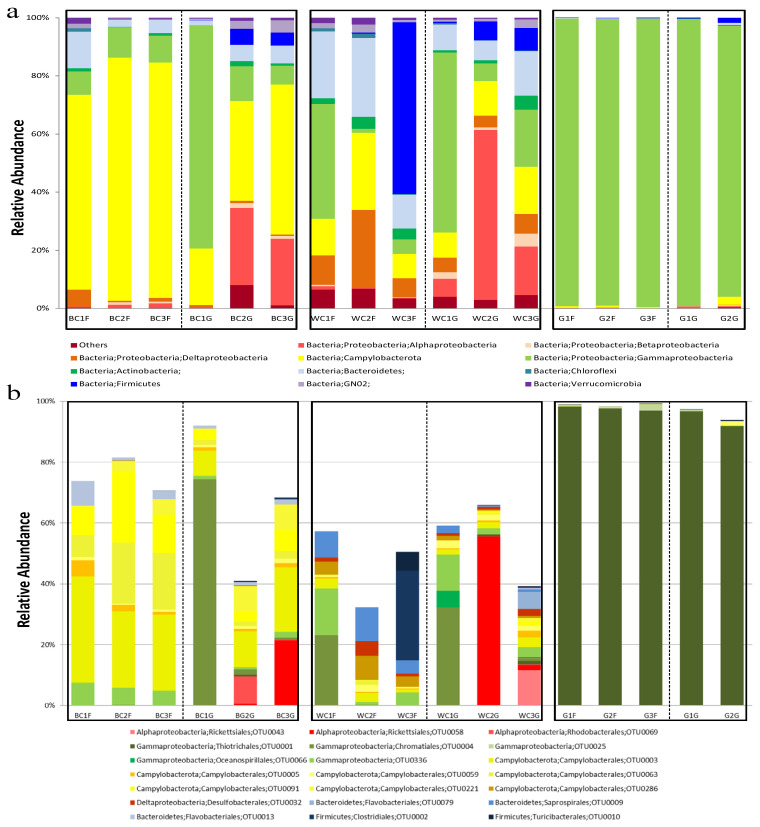
Bacterial communities based on relative abundance of ASVs from 16S rRNA gene amplicon sequences at class (**a**) and species (**b**) levels in the foot (F) and glands (G) of hydrothermal snails. In panel (**b**), the uncolored portion of each bar represents the cumulative abundance of taxa that could not be classified to the species level. BC, *Chrysomallon squamiferum* (black scaly); WC, *Chrysomallon squamiferum* (white scaly); G, *Gigantopelta aegis*.

**Figure 3 biology-14-00954-f003:**
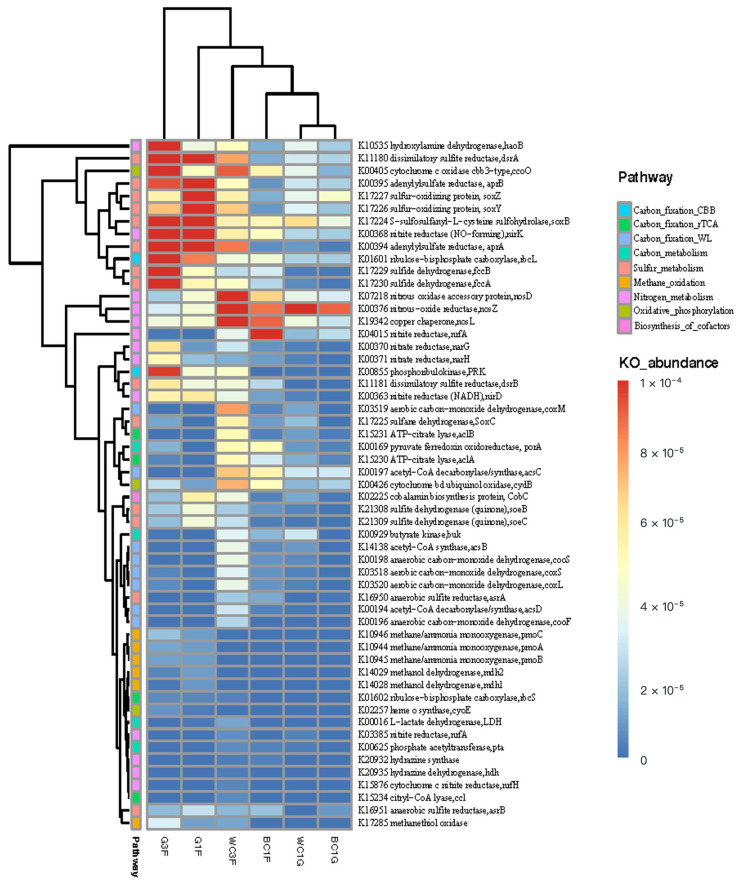
Heatmap and clustering of functional capacity profiles showing different enrichment in hydrothermal snails by metagenomic sequencing analysis. BC, *Chrysomallon squamiferum* (black scaly); WC, *Chrysomallon squamiferum* (white scaly); G, *Gigantopelta aegis*; F, foot; G, glands.

**Figure 4 biology-14-00954-f004:**
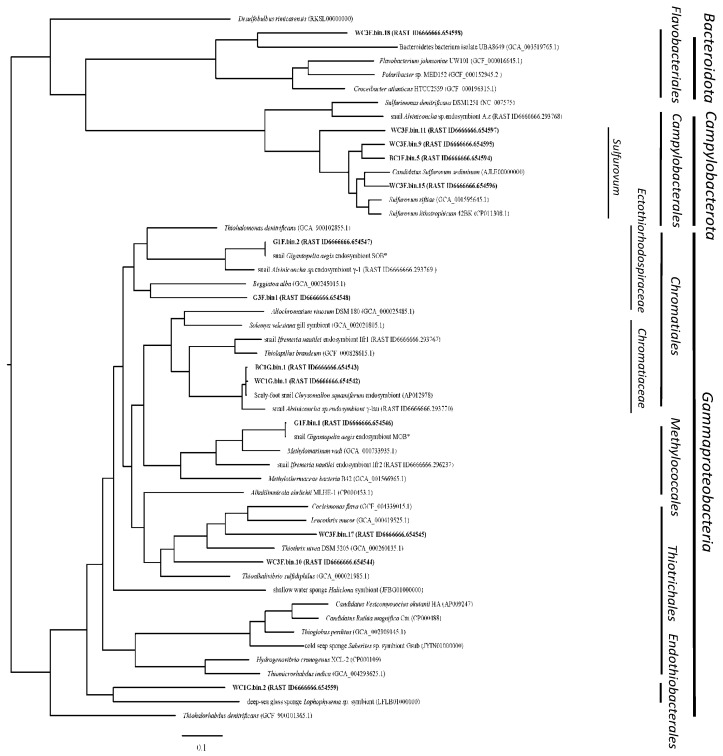
Phylogenomic tree of snail symbiont MAGs based on 120 marker genes, compared with other typical species and animal symbionts. * Indicates the same species as previously reported.

**Figure 5 biology-14-00954-f005:**
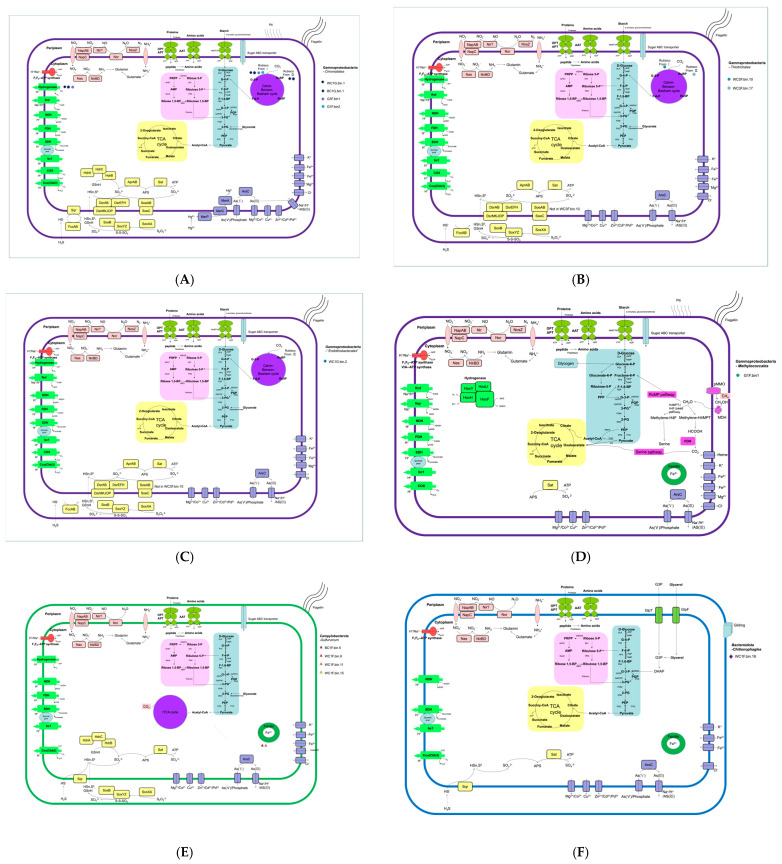
The major metabolisms of symbionts (13 MAGs) of orders Chromatiales (**A**) and Thiotrichales (**B**), “Endothiobacterales” (**C**) and Methylococcales (**D**) in Gammaproteobacteria, and Campylobacterota (**E**) and Bacteroidota (**F**) from the hydrothermal snails *Chrysomallon squamiferum* and *Gigantopelta aegis*.

**Figure 6 biology-14-00954-f006:**
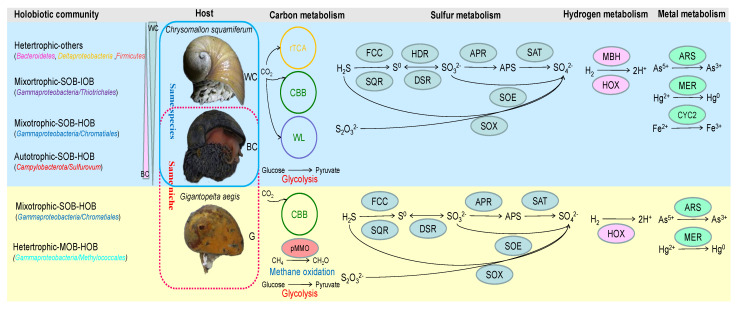
The schematic diagram showing the differences among the holobionts of *Chrysomallon squamiferum* and *Gigantopelta aegis*, in live style and symbiont metabolisms.

**Table 1 biology-14-00954-t001:** 16S rRNA amplicon sequence information and diversity estimates for 17 samples from *C. squamiferum* (BC, WC) and *G. aegis* (G).

Species	The Black Scaly *Chrysomallon squamiferum* (BC)						The White Scaly *Chrysomallon squamiferum* (WC)						*Gigantopelta aegis* (G)				
Samples	Scale Foot			Gland			Scale Foot			Gland			Foot			Gland	
**ID**	BC1F	BC2F	BC3F	BC1G	BC2G	BC3G	WC1F	WC2F	WC3F	WC1G	WC2G	WC3G	G1F	G2F	G3F	G1G	G2G
**Effective sequence tags**	83,413	41,705	48,417	102,478	86,262	77,152	72,329	15,542	71,883	72,329	58,503	65,641	95,143	89,073	84,091	101,179	100,557
**ASV**	187	284	326	206	417	264	291	197	323	469	344	336	98	145	99	126	204
**Shannon (H’)**	2.67	2.45	2.77	1.32	3.40	3.27	3.39	3.68	3.38	3.45	2.81	4.60	0.14	0.19	0.19	0.25	0.84
**Simpson**	0.15	0.16	0.13	0.56	0.05	0.11	0.09	0.04	0.10	0.13	0.31	0.03	0.97	0.95	0.94	0.94	0.84
**Chao1**	202.79	387.46	475.24	226.04	458.17	277.00	340.40	230.91	395.55	508.00	378.50	493.50	152.09	224.69	147.46	140.25	246.86
**Ace**	206.55	386.90	546.24	233.05	541.17	273.87	318.05	232.01	382.70	500.82	358.77	504.86	143.58	201.53	171.82	134.41	224.84

**Table 2 biology-14-00954-t002:** Comparison of genomic features among hydrothermal snail symbionts (13 MAGs).

Taxonomy	*Gammaproteobacteria*								*Campylobacterota*				*Bacteroidota*
	*Chromatiales*				*Thiotrichales*		*Methy-* *Lococcales*	“*Endothio-* *Bacterales*”	*Campylobacterales* */Sulfurovum*				*Chitin-* *Ophagales*
**MAGs**	WC1G. bin.1	BC1G. bin.1	G3F. bin1	G1F. bin2	WC3F. bin.10	WC3F. bin.17	G1F. bin.1	WC1G. bin.2	BC1F. bin.5	WC3F. bin.9	WC3F. bin.15	WC3F. bin.11	WC3F. bin.18
**Completeness**	97.59	99.74	92.76	98.55	80.28	80.16	98.13	88.48	82.77	87.89	85.38	89.95	95.32
**Genome size (bp)**	2,472,218	2,782,074	4,909,376	3,525,970	3,222,582	2,694,131	2,407,989	2,093,704	1,424,152	1,383,753	1,614,001	2,030,818	3,087,810
**GC (%)**	65.54	64.89	61.05	40.24	53.60	42.37	45.43	57.7	44.77	46.10	38.91	31.71	29.11
**No. protein coding gene**	2468	2668	4289	3256	3031	2369	2409	2187	1478	1438	1640	1906	2503
**Coding density (%)**	89.50	89.66	76.25	85.74	77.26	76.37	88.36	86.88	79.77	88.01	83.29	77.03	76.98
**Carbon Fixation**													
**CBB** **(form I)**	+	+	-	-	+	-	-	-	-	-	-	-	-
**CBB** **(form II)**	+	+	+	+	-	+	-	+	-	-	-	-	-
**rTCA**	-	-	-	-	-	-	-	-	+	+	+	+	-
**Sulfur oxidation**													
**SoxBAZYX**	+	+	+	+	+	+	-	+	-	-	-	+	-
**SoxCDYZ**	-	-	-	-	-	-	-	-	+	+	+	+	-
**Sqr**	+	+	+	+	+	+	-	+	+	+	+	+	+
**Fcc**	-	+	+	+	+	+	-	+	-	-	-	-	-
**HdrABC**	+	+	-	-	-	-	-	-	-	-	-	-	-
**DsrAB**	+	+	+	+	+	+	-	+	-	-	-	-	-
**AprAB**	+	+	+	-	+	+	-	+	-	-	-	-	-
**sat**	+	+	+	+	+	+	+	+	+	+	+	+	+
**SoeABC**	+	+	-	+	+	-	-	-	-	-	-	-	-
**Hydrogen oxidation**													
**MBHL**	+	+	+	-	-	+	-	-	-	-	-	-	-
**Hox**	+	-	+	-	+	-	+	-	-	-	-	-	-
**Methano oxidation**													
**pMMO**	-	-	-	-	-	-	+	-	-	-	-	-	-
**CO oxidation**													
**Coo**	-	-	-	-	-	-	-	-	-	-	-	-	-
**Metal utilization and resistance**													
**Iron oxidase Cyc2**	-	-	-	-	+	+	-	-	-	+	-	-	-
**Iron reduction genes**	-	-	-	-	-	-	-	-	-	-	-	-	-
**Iron storage genes**	-	-	-	-	-	-	+	+	+	-	+	-	+
**Arsenite oxidation genes**	-	-	-	-	-	-	-	-	-	-	-	-	-
**ArsC**	+	+	+	+	+	+	+	+	+	+	+	+	+
**MerA**	+	+	+	+	+	-	-	-	-	-	-	-	-
**Copper resistance**	+	+	+	+	+	+	+	+	+	+	+	+	+
**Oxygen respiration**													
**Cox**	+	+	+	+	-	-	+	+	-	-	-	-	+
**Cco**	+	+	+	-	+	+	-	-	+	+	+	+	+
**Glc**	**+**	**+**	**+**	**-**	**+**	**+**	**-**	**-**	**-**	**-**	**-**	**-**	**-**
**Nitrate and nitrite ammonification**													
**NapAB**	**+**	**+**	**+**	**+**	**+**	**+**	**+**	**+**	**-**	**+**	**+**	**+**	**+**
**NasA**	**+**	**+**	**+**	**+**	**+**	**-**	**+**	**+**	**+**	**-**	**+**	**-**	**+**
**NirBD**	**+**	**+**	**+**	**+**	**+**	**-**	**+**	**+**	**+**	**-**	**+**	**-**	**+**
**Electron transport chain**													
**ATP synthase**	F-type	F-type	F-type; V-type	F-type	F-type	F-type	F-type; V-type	F-type	F-type	F-type	F-type	F-type	F-type
**Nuo**	+	+	+	**+**	+	+	-	**+**	**+**	**+**	**+**	**+**	**+**
**Fdh**	+	+	+	**-**	-	-	+	**-**	**-**	**-**	**-**	**-**	**-**
**Sdh**	+	+	+	**+**	+	+	+	**+**	**+**	**+**	**+**	**+**	**+**
**Nqr**	-	-	-	**-**	-	-	+	**-**	**-**	**-**	**-**	**-**	**-**
**Rnf**	+	+	+	**+**	-	+	+	**+**	**-**	**-**	**-**	**-**	**-**
**Motility**													
**Flagellum**	+	+	**-**	**+**	**+**	**-**	+	**-**	**-**	**-**	**-**	**-**	**-**
**Pili**	+	+	**+**	**+**	**+**	**+**	+	**+**	**-**	**-**	**-**	**-**	**-**
**Gliding**	-	**-**	**-**	**-**	**-**	**-**	-	**-**	**-**	**-**	**-**	**-**	**+**
**Vitamin and cofactor**													
**Thiamine (Vitamin B1)**	+	**+**	**+**	+	**-**	**+**	+	**+**	**+**	**+**	**+**	**+**	**-**
**Riboflavin (Vitamin B2)**	+	**+**	**+**	**-**	**-**	**+**	-	**+**	**+**	**+**	**+**	**+**	**+**
**pyridoxine (Vitamin B6)**	+	**+**	**-**	+	**-**	**+**	+	**-**	**+**	**+**	**-**	**+**	**-**
**Biotin (Vitamin B7)**	+	**+**	**+**	+	**+**	**-**	+	**+**	**+**	**-**	**-**	**+**	**-**
**Folic acid (Vitamin B9)**	+	**+**	**+**	+	**+**	**-**	+	**+**	**+**	**+**	**+**	**+**	**+**
**Cobalamin (vitamin B12)**	-	**-**	**-**	+	**+**	**-**	+	**+**	**-**	**-**	**-**	**-**	**-**

**Table 3 biology-14-00954-t003:** Diverse symbiotic metabolisms in hydrothermal snails belong to the family Provannidae and the family Peltospiridae, class Gastropoda in the phylum Mollusca. SOB, sulfur-oxidizing bacteria; HOB, hydrogen-oxidizing bacteria; MOB, methane-oxidizing bacteria; HB, heterotrophic bacteria.

Host	Habitat	Symbiont Location	Symbiont Type	Main Symbiont Community	Refs
***Alviniconcha* spp.** **/Provannidae**	SWIR, CR, Indian Ocean; Western Pacific Ocean	Intracellular; Gill;	SOB; HOB;	*Sulfurimonas*/*Sulfurovum*, *Helicobacteraceae*, *Campylobacterales*, *Campylobacterota*	[7,71]
			SOB; HOB;	*Ectothiorhodospiraceae*, *Chromatiales*, *Gammaproteobacteria* *Chromatiaceae*, *Chromatiales*, *Gammaproteobacteria*	
***Gigantopelta* spp.** **/Peltospiridae**	SWIR, Indian Ocean; Southern Ocean	Intracellular; Gland;	SOB; MOB	*Ectothiorhodospiraceae*, *Chromatiales*, *Gammaproteobacteria* *Methylococcaceae*, *Methylococcales*, *Gammaproteobacteria*	[13] This study
** *Chrysomallon squamiferum* ** **/Peltospiridae**	SWIR, CIR, CR, Indian Ocean	Scale;	SOB; HOB; HB	*Sulfurovum*, *Helicobacteraceae*, *Campylobacterales*, *Campylobacterota* *Thiotrichaceae*, *Thiotrichales*, *Gammaproteobacteria* *Flavobacteriales*, *Bacteroidota*	[16,36] This study
		Intracellular; Gland;	SOB; HOB;	*Chromatiaceae*, *Chromatiales*, *Gammaproteobacteria* *Thiotrichaceae*, *Thiotrichales*, *Gammaproteobacteria* “*Endothiobacterales*”, *Gammaproteobacteria*	

## Data Availability

The 16S rRNA genes and metagenomics sequencing data that support the findings of this study have been deposited in the CNSA (https://db.cngb.org/cnsa/) of CNGBdb with accession code CNP0001245. The assembled and annotated symbiont genomes are also publicly available on the RAST server (http://rast.theseed.org/) using the guest login with IDs 6666666.654542-6666666.654548, 6666666.654559, and 6666666.654594-6666666.654598.

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
