# Peer review of "Host Shaping Associated Microbiota in Hydrothermal Vent Snails from the Indian Ocean Ridge"

_biology, 2025, doi:10.3390/biology14080954_

Round 1
Reviewer 1 Report
Comments and Suggestions for Authors
Dear authors,
This manuscript provides novel insight into host-microbe interactions in Indian Ocean hydrothermal vent snails. The data are well-analyzed, and the identification of distinct symbiont communities between three snail species is particularly compelling. However, English should be improved for exact demonstration of the contents. Additionally, you should use the term ASV rather than OTU and need to update scientific names of the bacterial phyla.
Sometimes, mis-spelling and not verified numbers are shown, hence, need to check MS thoroughly.
Figure titles should be described under the figures in supplementary file.
You can found detailed in attached file.

Author Response
Dear Reviewer,
we sincerely appreciate your insightful comments and suggestions, which have greatly improved our manuscript. We have carefully considered your feedback and have revised the manuscript accordingly . Thank you for your time and expertise.
Best regards,
Xiang ZENG
Reviewer 2 Report
Comments and Suggestions for Authors
Review for biology-3702585: Host shaping associated microbiota in hydrothermal vent snails from the Indian Ocean ridge
General comments
This study by Zeng et al. investigates the esophageal gland and foot microbiome of two sympatric hydrothermal vent snails, Gigantopelta aegis and Chrysomallon squamiferum, using 16S rRNA amplicon sequencing and metagenomics. The authors find notable differences in microbial composition and metabolic potential between host species, snail ecotypes as well as tissue types, suggesting that both host specificity and environmental factors influence the microbiomes of these snail species. I found this study very interesting and comprehensive, and the data are relevant to the deep-sea research community. However, I have several suggestions for improvement and also some concerns that I would like the authors to address in a revision. There are some factual mistakes in the manuscript, and I would urge the authors to pay more attention to wording. The results and methods sections need some more clarity. In particular, the selection of samples for sequencing needs to be explained a bit more as the esophageal gland samples from G. aegis were surprisingly not included in the analyses. I also think the manuscript would benefit from a deeper integration of the collected environmental parameters and observed differences in microbiome composition. Since one main knowledge gap the authors highlighted was the interaction between ecological niche and holobiont I was expecting some kind of redundancy analysis to understand how environmental parameters influence microbiome composition and metabolic potential. I would advise the authors to add these tests to the manuscript. I give some more detailed recommendations below.
Introduction
- L84: Maybe point out here that this snail does not have any scales on its foot…
- I think the introduction could benefit from some more detailed description of the ecological niches the two snail species occupy as well as some information on the symbiont transmission system
Methods
- L108-129: This section contains a lot of information that is not related to sample collection. To make it easier for the reader please describe the microscopy and environmental fluid analyses in separate sections.
- L109: Gigantopelta does not contain scales on its foot. Please correct this mistake by rephrasing the sentence, e.g., “Individuals of Gigantopelta aegis (abbreviated as G) and the black scaly-foot snail Chrysomallon squamiferum(abbreviated as BC) were collected…”
- L118: Change “scaly foot” to “foot” (since aegis does not contain any scales)
- L126-129: How and where was the fluid collected? Please add more details on the collection/measurement of environmental parameters. Since you have information on multiple environmental parameters it would be worthwhile to conduct a redundancy analysis to test for a relationship between vent environment and microbial community composition / metabolic potential. Since the niche-holobiont interaction was one main knowledge gap you highlighted in the introduction I think this analysis is essential and would strengthen the manuscript significantly. Please add these tests and discuss the results accordingly
- L130 and L151: Please mention in these sections how many and which samples exactly were analyzed
- L142: Since you use the unoise3 program and thus cluster sequences by 100% identity I would change OTU to ASV here and in the following text
- L149: Please change MAG abundance to ASV abundance
- L171-172: How did you distinguish host genes from symbiont genes here? Was this part of the manual screening? Given the amount of data, I think some bioinformatic processing would need to be applied to accurately filter out host contaminants
- L174-176: Move this sentence to the end of the section to list the analysis steps chronologically and make it easier for the reader to follow the workflow
Results
- Results in Tables 1 and 2 are very difficult to read as numbers are very closely spaced and/or are spread across lines. Maybe use a smaller font?
- L267-304: This section is very complex and difficult to read. Maybe try to rephrase this and just point out the broader patterns across species/ecotypes instead of noting patterns for each sequenced individual.
- L326-327: Why were two foot samples for aegis analyzed but no gland sample? Based on the 16S rRNA results the communities look very similar between tissue types in G. aegis but I was still expecting to see metagenomic sequencing of both tissue types. I think the choice of samples for sequencing needs to be explained and justified in much more detail.
- L351-353: It would be interesting to see how the aegis gland sample compares to the rest. Looking at the heatmap the samples cluster by tissue type for C. squamiferum based on metabolic potential but cluster by ecotype (WC versus BC) based on taxonomic composition
- L379-596: This section is very long and information-rich. It will be difficult for readers to remember most of this information. I would advise to condense this section by highlighting the main features for the different MAGs and put the full description into the supplement
- Figure 2b: It looks like there is a mistake in the plotting of the data. These should be relative abundances so bars should go up to 100% but instead the bar heights differ. Did you plot read counts instead maybe?
Discussion
- L617-618: Be careful with this statement. There are quite a few studies on hydrothermal vents now.
- L635: Gigantopelta and Chrysomallon belong to the family Peltospiridae, not Provannidae
- L653-654: The broad distribution of squamiferum could also be related to differences in dispersal potential. Please moderate this statement and indicate that the metabolic versatility of its symbiont might just be one driver for the wide-spread distribution
- L698-699: As far as I know vertical transmission has not been unequivocally proven in this system. Please moderate this statement. Based on Lan et al. (2022), squamiferum symbionts likely employ a mixed transmission mode, though further analyses are needed to confirm this
- L706-731: This section needs more work as I am not completely sure what the authors want to point out here. With the currently available data, it is not possible to link species distribution to the diversity of their symbionts. There are multiple other factors that influence species distribution, such as dispersal potential, habitat conditions (other than symbiont availability), inter-species competition etc.
- L716-717: Alviniconcha boucheti and marisindica can also host Sulfurovum as symbionts and they are currently believed to be endosymbionts
- Table 3: These snail species belong to very divergent subclasses. Only Alvinivoncha and Ifremeria are part of the family Provannidae. I would remove this table as it is not necessary
- It would be great if you could include some more discussion on the differences in microbial composition and metabolic potential that you observed between tissue types within each species
Author Response
Dear Reviewer,
we sincerely appreciate your insightful comments and suggestions, which have greatly improved our manuscript. We have carefully considered your feedback and have revised the manuscript accordingly. Thank you for your time and expertise.
Best regards,
Xiang ZENG

Round 2
Reviewer 2 Report
Comments and Suggestions for Authors
I thank the authors very much for addressing my comments. The manuscript is much improved and I am almost satisfied with the current revisions. There are just a couple of minor things that still need to be corrected:
- The use of the word scaly/scale needs to be fixed in the following lines:
- L24-25: Replace “snail scales” with “snail foot”
- L32-33: Replace “scales” with “foot”
- L217: remove “scaly”
- L315: remove “scaly”
- L350-351: remove “scaly”
- L282-302: Since the microbiome composition has not been described at this point, I would move this section to line 382 where you first explain the differences in community composition between snail species
- Line 319 should read into “denoised ASVs (also called zero-radius OTUs, or ZOTUs (Operational Taxonomic Units))”. The term zero-radius OTU is a synonym for ASV and is correct here. In the Abbreviations table ASV should correspond to Amplicon Sequence Variant (not Operational Taxonomic Unit)
- L414ff: I appreciate that the authors added this information as I had recommended. However, optimally you would mention this already in the methods to signal to readers that no gland tissues from G. aegis were used for metagenomics. Currently, the methods make it sound like all samples passed the quality checks so mentioning here that the gland samples for G. aegis actually failed is a bit confusing.
- L350: I think the values for glands and foots are mixed up in this sentence. This should read: “Likewise, the mean Chao1 index 350 in glands (341.04) was higher than the mean Chao1 index in the foot (284.07).
- L914: change “ectosymbiont” to “symbiont”. As I mentioned in my previous review the Alviniconcha symbionts are currently assumed to be all endosymbionts (not ectosymbionts)
- L927-929: Remove this section. I am not aware of any study that looked at microbiome composition in digestive glands of Alviniconcha. The two cited references did not. Please also remove this information from Table 3
- Table 3: Remove the row containing information about gland symbiont communities in Alviniconcha. Alviniconcha snails are only known to harbor symbionts in the gill tissue
There are still some grammatical and spelling errors present in the manuscript. I advise you to proof-read the text again
Author Response
Dear Reviewer,
Thank you very much for your feedback. We have carefully addressed your remaining concerns, made the necessary corrections, and thoroughly checked the entire document for minor errors. The specific revisions have been highlighted in purple for your review.
Below, we respond to each of your comments individually:
I thank the authors very much for addressing my comments. The manuscript is much improved and I am almost satisfied with the current revisions. There are just a couple of minor things that still need to be corrected:
- The use of the word scaly/scale needs to be fixed in the following lines:
- L24-25: Replace “snail scales” with “snail foot”
- L32-33: Replace “scales” with “foot”
- L217: remove “scaly”
- L315: remove “scaly”
- L350-351: remove “scaly”
R: Thank you. We modified them.
- L282-302: Since the microbiome composition has not been described at this point, I would move this section to line 382 where you first explain the differences in community composition between snail species
R: Thank you. We moved them to 3.2.1.
- Line 319 should read into “denoised ASVs (also called zero-radius OTUs, or ZOTUs (Operational Taxonomic Units))”. The term zero-radius OTU is a synonym for ASV and is correct here. In the Abbreviations table ASV should correspond to Amplicon Sequence Variant (not Operational Taxonomic Unit)
R: Thank you. We modified the faults.
- L414ff: I appreciate that the authors added this information as I had recommended. However, optimally you would mention this already in the methods to signal to readers that no gland tissues from G. aegis were used for metagenomics. Currently, the methods make it sound like all samples passed the quality checks so mentioning here that the gland samples for G. aegis actually failed is a bit confusing.
R: Thank you. We added description in the method 2.3 and result 3.2.2.
- L350: I think the values for glands and foots are mixed up in this sentence. This should read: “Likewise, the mean Chao1 index 350 in glands (341.04) was higher than the mean Chao1 index in the foot (284.07).
R: Thank you. We modified it.
- L914: change “ectosymbiont” to “symbiont”. As I mentioned in my previous review the Alviniconcha symbionts are currently assumed to be all endosymbionts (not ectosymbionts)
R: Thank you. We modified it.
- L927-929: Remove this section. I am not aware of any study that looked at microbiome composition in digestive glands of Alviniconcha. The two cited references did not. Please also remove this information from Table 3
R: Thank you. We modified it.
- Table 3: Remove the row containing information about gland symbiont communities in Alviniconcha. Alviniconcha snails are only known to harbor symbionts in the gill tissue
R: Thank you for comments. We deleted it.
Best regards,
Xiang ZENG